# Antitubercular Activity of 7-Methyljuglone-Loaded Poly-(Lactide Co-Glycolide) Nanoparticles

**DOI:** 10.3390/pharmaceutics16111477

**Published:** 2024-11-20

**Authors:** Bianca Diedericks, Anna-Mari Kok, Vusani Mandiwana, Bhavna Gowan Gordhan, Bavesh Davandra Kana, Suprakas Sinha Ray, Namrita Lall

**Affiliations:** 1Department of Plant and Soil Sciences, University of Pretoria, Pretoria 0002, South Africa; 17023409@tuks.co.za (B.D.); annamarikok@gmail.com (A.-M.K.); 2South African International Maritime Institute (SAIMI), Nelson Mandela University, Gqeberha 6019, South Africa; 3Centre for Nanostructures and Advanced Materials, DSI-CSIR Nanotechnology Innovation Centre, Council for Scientific and Industrial Research, Pretoria 0001, South Africa; vmandiwana@csir.co.za (V.M.); rsuprakas@csir.co.za (S.S.R.); 4National Health Laboratory Service, School of Pathology, Faculty of Health Science, University of the Witwatersrand, Johannesburg 2000, South Africa; bhavna.gordhan@nhls.ac.za (B.G.G.); bavesh.kana@wits.ac.za (B.D.K.); 5Department of Chemical Sciences, University of Johannesburg, Doornfontein 2028, South Africa; 6School of Natural Resources, University of Missouri, Columbia, MO 65211, USA; 7College of Pharmacy, JSS Academy of Higher Education and Research, Mysuru 570015, Karnataka, India; 8Bio-Tech R&D Institute, University of the West Indies, Kingston 770, Jamaica

**Keywords:** tuberculosis, antimycobacterial, cytotoxicity, 7-methyljuglone, poly-(lactide-co-glycolide), nanoparticle

## Abstract

Background/Objectives: Loading of natural products into poly-(lactide-co-glycolic) acid (PLGA) nanoparticles as drug delivery systems for the treatment of diseases, such as tuberculosis (TB), has been widely explored. The current study investigated the use of PLGA nanoparticles with 7-methyljuglone (7-MJ), an active pure compound, isolated from the roots of *Euclea natalensis* A. DC. Methods: 7-MJ as well as its respective PLGA nanoparticles were tested for their antimycobacterial activity against *Mycobacterium smegmatis* (*M. smegmatis*), drug-susceptible *Mycobacterium tuberculosis* (*M. tuberculosis*) (H37Rv), and multi-drug-resistant *M. tuberculosis* (MDR11). The cytotoxicity of 7-MJ as well as its respective PLGA nanoparticles were tested for their cytotoxic effect against differentiated human histiocytic lymphoma (U937) cells. Engulfment studies were also conducted to determine whether the PLGA nanoparticles are taken up by differentiated U937 cells. Results: 7-MJ has been shown to have a minimum inhibitory concentration (MIC) value of 1.6 µg/mL against *M. smegmatis* and multi-drug-resistant *M. tuberculosis* and 0.4 µg/mL against drug-susceptible *M. tuberculosis*. Whilst promising, 7-MJ was associated with cytotoxicity, with a fifty percent inhibition concentration (IC_50_) of 3.25 µg/mL on differentiated U937 cells. In order to lower the cytotoxic potential, 7-MJ was loaded into PLGA nanoparticles. The 7-MJ PLGA nanoparticles showed an 80-fold decrease in cytotoxic activity compared to free 7-MJ, and the loaded nanoparticles were successfully taken up by differentiated macrophage-like U937 cells. Conclusions: The results of this study suggested the possibility of improved delivery during TB therapy via the use of PLGA nanoparticles.

## 1. Introduction

In several fields of medicine, nanoparticles have been successfully applied in therapeutic strategies for various conditions, one of which includes infectious diseases. Regarding the treatment of tuberculosis (TB), nanoparticles are useful as delivery systems for known or novel antitubercular (anti-TB) drugs, an approach that reduces side effects associated with drug dose and toxicity [1]. Nanoparticles are colloidal, submicron sized particles measuring less than 1000 nm [2]. Various materials can be used to synthesize either organic (liposomes, solid-lipid nanoparticles, nano-emulsions, micelles, and polymeric nanoparticles) or inorganic nanoparticles (silver (Ag), gold (Au), gallium (Ga), and copper (Cu) nanoparticles) [3,4]. Both inorganic and organic nanoparticles can be synthesized using various methods, resulting in nanoparticles of different sizes and structures, with consequent effects on the final physical and chemical properties [5].

Nanoparticles offer several key advantages as drug delivery systems. They can improve the aqueous solubility of poorly soluble drugs. Nanoparticles can also carry both hydrophobic and hydrophilic drugs. Additionally, they protect drugs from degradation. Finally, nanoparticles allow controlled release, which can reduce the frequency of administration and the required dose [1,6,7]. Reducing the dose of a drug required to achieve a certain therapeutic effect will in turn help to reduce any toxicity resulting in fewer side effects [8]. The toxicity of nanoparticles is influenced by the chemical composition of the materials used in their synthesis; therefore, nanoparticles can be modified to ensure they carry no harmful risk to human health [9].

Poly (lactic-co-glycolic acid) (PLGA) is a material that is approved by the Food and Drug Administration (FDA) for the synthesis of innocuous nanoparticles. This biodegradable functional polymer is produced by polymerizing lactic acid (LA) and glycolic acid (GA). It is extensively applied in pharmaceuticals and medical engineering materials, due to its biocompatible nature, plasticity, and non-toxic properties [10]. PLGA nanoparticles have been studied for improving TB treatment. One key advantage is their ability to accumulate in *Mycobacterium tuberculosis* (*M. tuberculosis*)-infected macrophages. These macrophages are the primary immune cells responsible for controlling the infection [11,12]. The accumulation of PLGA nanoparticles loaded with anti-TB drugs in the host cell enhances the drug activity at the infection site and reduces systemic toxicity [13]. PLGA nanoparticles loaded with anti-TB drugs provide sustainable release of drugs over extended periods, which ensures a continuous therapeutic drug concentration at the infection site, thereby improving treatment efficacy and reducing the frequency of dosing [14]. Some anti-TB drugs are vulnerable to degradation or unstable in the body; therefore, encapsulating them into PLGA nanoparticles may provide protection thus preserving their effectiveness during storage outside the body and biodistribution in the body [15].

7-Methyljuglone (7-MJ), a pure compound isolated from the roots of *Euclea natalensis* A. DC, has previously been shown to be a promising additive to the current TB treatment regimen as it has proven activity against both pathogenic and non-pathogenic mycobacterial species, such as *M. tuberculosis* and *Mycobacterium smegmatis* (*M. smegmatis*), respectively [16,17,18]. 7-MJ, however, exhibits cytotoxicity against several cell lines, including human histiocytic lymphoma (U937), mouse macrophage (J774A.1), and umbilical vein endothelial cells (HUVEC) [19,20,21].

The aim of this study was to characterize and evaluate the stability of the 7-MJ PLGA nanoparticles and assess whether the nanoparticles had enhanced activity compared to the pure compound. 7-MJ PLGA nanoparticles were tested for their cytotoxic effect on differentiated U937 cells and for their antimycobacterial activity against both *M. smegmatis* and *M. tuberculosis*. Furthermore, the 7-MJ PLGA nanoparticles were quantified within differentiated U937 macrophage-like cells; this will in turn indicate whether the drug will be able to treat the infection within the infected cells.

## 2. Materials and Methods

### 2.1. Chemicals, Reagents, and Pure Compound

All solvents and reagents utilized during this study were obtained from Sigma Aldrich (St. Louis, MO, USA) and were of analytical grade. The 7-MJ pure compound was obtained from the extract library of Prof Namrita Lall at the Department of Plant and Soil Sciences, University of Pretoria. The full extraction and purification method used to obtain the 7-MJ is explained by Van der Kooy et al. (2006) [18].

### 2.2. PLGA Nanoparticle Synthesis and Characterization

Three PLGA nanoformulations were prepared; blank nanoparticles, 7-MJ nanoparticles, and 7-MJ + Rodamine B (RhB) nanoparticles. The PLGA nanoparticles were prepared through a nanoprecipitation technique. To prepare the blank nanoparticles and 7-MJ nanoparticles, an oil-in-water (O/W) emulsion was used. For the 7-MJ + RhB nanoparticles, a water-in-oil-in-water (W/O/W) emulsion was prepared. For the blank nanoparticles, the oil phase consisted of 50 mg of PLGA (50:50 → LA:GA) dissolved in 5 mL of acetone. For the nanoparticles containing 7-MJ (7-MJ and 7-MJ + RhB), the oil phase included 5 mg of 7-MJ and 50 mg of PLGA (50:50 → LA:GA) dissolved in 5 mL of acetone. For the internal aqueous phase of the 7-MJ + RhB nanoparticles, 1 mL of 2% RhB was prepared. The RhB solution was added dropwise to the oil phase. The external aqueous continuous phase of each formulation constituted 1% polyvinyl alcohol (PVA). The addition of the above-mentioned oil phase was conducted in a dropwise manner to the external aqueous phase for each formulation. After the addition, one drop of 2% Tween 80 was added to each formulation, while stirring on a magnetic stirring plate. The drop of Tween 80 functioned as an emulsifier with the purpose of breaking the surface tension between the oil and water phase. For each of the nanoformulations, the heterogeneous solution was stirred for approximately 10 min. The solution was then blended by means of a Silverson L4R high-speed homogenizer (Silverson Machines Limited, Buckinghamshire, UK) at 10,000 rpm for 10 min on ice. Then, the emulsion was left to stir for 24 h. The resultant emulsion was centrifuged for 30 min at 10,000 rpm to form a nanoparticle pellet. The supernatant was discarded, and the pellet was resuspended in 5 mL of 1% Trehalose. The resuspended pellet was rapidly frozen with liquid nitrogen and freeze-dried over a period of 2 days to obtain freeze-dried PLGA nanoformulations (blank nanoparticles, 7-MJ nanoparticles and 7-MJ + RhB nanoparticles). Illustrative diagrams of the synthesis of the PLGA nanoparticles are provided in Appendix A).

#### 2.2.1. Dynamic Light Scattering

##### Particle Diameter and Polydispersity Index

The diameter of the particle and size distribution specified as the polydispersity index (PdI) were assessed via dynamic laser scattering (DLS) utilizing a Malvern Zetasizer Nano ZS (Malvern Instruments, Worcestershire, UK). To initiate the characterization, 1 mg of each freeze-dried PLGA nanoparticle sample was resuspended in 1 mL distilled water (dH_2_O), added to a U-shaped Zeta cell, and analyzed. The background signal was reduced by using dH_2_O as a blank. Each sample was measured in triplicate.

##### Surface Charge

The Malvern Zetasizer Nano ZS (Malvern Instruments, Worcestershire, UK) at a pH of 6.8 was used to determine the zeta potential of the nanoparticles in suspension. The Laser Doppler Velocity principle was used by the instrument to calculate the nanoparticles’ net charge at the surface by establishing electrophoretic mobility. Similar to the above-mentioned analysis, 1 mg of each freeze-dried PLGA nanoparticle sample was resuspended in 1mL dH_2_O and transferred to a Zeta cell. Each sample was measured in triplicate.

#### 2.2.2. Potential of Hydrogen

Potential of hydrogen (pH) readings of the nanoparticles were taken using an Orion 3-Star Benchtop pH Meter (Thermo Fisher Scientific, Waltham, MA, USA). For a period of 6 months, once a month, 1 mg of each freeze-dried PLGA nanoparticle sample was resuspended in 1 mL dH_2_O and their pH readings were taken.

#### 2.2.3. Fourier Transform Infrared Spectroscopy

All the synthesized nanoparticles’ infrared spectra were investigated using Fourier transform infrared spectroscopy (FTIR) (PerkinElmer Spectrum 100 FTIR spectrometer, PerkinElmer, Midrand, South Africa) to characterize functional groups and study possible interactions and bonding patterns between the PLGA, the pure compound, and the fluorescent dye. To eliminate background noise, an empty read (no sample) was used as a blank. The infrared range of 650–4000 cm^−1^ was used to detect the percentage (%) transmittance.

#### 2.2.4. Method for Drug Analysis

The drug content in the 7-MJ nanoparticles and the 7-MJ + RhB PLGA nanoparticles was measured via ultraviolet–visible (UV-Vis) spectroscopy. The UV5Bio spectrophotometer (Mettler Toledo, Columbus, OH, USA) with wavelengths between 190 and 800 nm was used to carry out the analyses. The broad wavelength scan was performed to identify the optimal wavelength for measuring the drug content of 7-MJ and RhB, respectively. The concentration range of the standards (7-MJ and RhB) was 100 to 1000 μg/mL.

##### Drug Loading Content (DLC) and Encapsulation Efficiency (EE) of the PLGA Nanoparticles

Drug loading content and encapsulation efficiency measures the success of loading the pure compound and fluorescent dye into the PLGA nanoparticle matrix and takes into consideration the final mass of the synthesized nanoparticles as well as the initial mass of the compound administered, respectively. These were evaluated according to the following formulae:$$DLC\;(\%)=\frac{Weight\;of\;drug\;in\;nanoparticle}{Weight\;of\;nanoparticle\;and\;drug\;}\times 10$$
$$EE\;(\%)=\frac{Weight\;of\;drug\;in\;nanoparticle}{Initial\;amount\;of\;drug\;taken\;for\;encapsulation\;}\times 100$$

#### 2.2.5. X-Ray Diffraction

X-ray diffraction analysis (XRD) was used to determine the crystallographic structure of each freeze-dried PLGA nanoparticle. The blank nanoparticles, 7-MJ nanoparticles, and 7-MJ + RhB nanoparticles were fixed on sample mounting stages using glass coverslips. Using a PANalytical X’Pert PRO (PANalytical, Almelo, The Netherlands), the crystalline structure of the nanoparticles was determined by illuminating the nanoparticles with monochromatized Cu Kα radiation (λ = 0.5 Å) between 5 °C and 90 °C (2θ) with a 0.026 °C step size. A current of 40 mA and a voltage of 45 kV were set as parameters.

#### 2.2.6. Stability Studies of PLGA Nanoparticles

##### Storage-Stability Studies

A storage-stability study was conducted for all the PLGA nanoparticles synthesized, by dissolving 1 mg of the nanoparticles into 1 mL of dH_2_O with repeated readings of DLS and pH taken once a month for a period of 6 months to ensure that the nanoparticles remain stable over time in their storage conditions. The PLGA nanoparticles were stored at −20 °C to preserve their stability and prevent degradation over time.

##### In Vitro Stability Within Various Biological Media

In vitro stability of the synthesized PLGA nanoparticles was evaluated in various buffer solutions and media throughout this study, which included pH buffer solutions at pH 4, 7, and 10, phosphate buffered saline (PBS), Roswell Park Memorial Institute (RPMI) 1640 media, RPMI 1640 media with 0.1% phorbol 12-myristate 13-acetate (PMA), 0.5% bovine serum albumin (BSA), 0.5% Cysteine, 5% sodium chloride (NaCl), distilled water (dH_2_O), and 7H9 media with Tween 80 and without Tween 80. The blank nanoparticles and the 7-MJ nanoparticles were added to the above-mentioned solutions at a 1:1 ratio with a final volume of 1 mL and were incubated at 37 °C. The 7-MJ + RhB nanoparticles were added to PBS, RPMI 1640 media, RPMI 1640 media with PMA, 5% NaCl and dH_2_O at a 1:1 ratio with a final volume of 1 mL, and were incubated at 37 °C. To confirm whether the nanoparticles were stable, their mean hydrodynamic diameter was read at 2, 24, 48, 72, 96, 120, and 144 h using a Malvern Zetasizer Nano ZS (Malvern Instruments, Worcestershire, UK).

### 2.3. In Vitro Antimycobacterial Activity on Mycobacterium smegmatis

The antimycobacterial activity of the pure compound (7-MJ), blank nanoparticles, and 7-MJ nanoparticles against *Mycobacterium smegmatis* were investigated using the Microtiter Alamar Blue assay (MABA) method, as defined by Franzblau et al. with alterations as indicated by Lall et al. [22,23]. The *M. smegmatis* were grown on Middlebrook 7H10 agar plates for 7 days before use in the MPBA assay. To each well of a sterile flat-bottom 96-well plate, 100 µL of Middlebrook 7H9 media (containing 0.4% glycerol and 0.5% Tween 80) was added. To the first rows of the plate, 100 µL of the pure compound, nanoparticle formulations as well as the positive control, ciprofloxacin, were added in triplicate. A two-fold serial dilution of each sample and ciprofloxacin was performed, which yielded a final concentration range from 1000 to 15.62 µg/mL for the nanoparticles and 0.08 to 5 µg/mL for ciprofloxacin and 7-MJ. Additional controls included an untreated bacterial negative control, an untreated 7H9 media control as well as solvent controls, 2.5% dH_2_O (nanoparticles), and 2.5% DMSO (7-MJ). A final volume of 200 µL was achieved via the addition of 100 µL of *Mycobacterium smegmatis* bacterial inoculum to all wells of the 96-well plate, excluding the media control wells. The concentration of bacterial inoculum was optimized according to McFarland’s standards. A 0.5 McFarland (OD600 = 0.085) was prepared and diluted to a concentration of 1.5 × 10^6^ CFU/mL. The plates were incubated for a period of 24 h at 37 °C in a CO_2_ incubator. Post-incubation, 20 µL of Presto Blue was added to all wells, followed by an additional incubation for 4 h at 37 °C. A conversion of the blue resazurin to a pink resorufin indicated cell viability, and the minimum inhibitory concentration (MIC) value was subsequently determined as the minimal concentration of sample/ciprofloxacin for which no color change occurred.

### 2.4. In Vitro Antimycobacterial Activity on Mycobacterium tuberculosis

The antimycobacterial activity of 7-MJ, blank nanoparticles, and 7-MJ nanoparticles against a drug-susceptible *M. tuberculosis* strain (H37Rv) and a multi-drug-resistant strain (MDR11) was investigated using the broth microdilution minimum inhibitory concentration method as previously described by Kana et al., with modifications [24]. To each well of a sterile round-bottom 96-well plate, 100 µL of Middlebrook 7H9 media (containing 0.4% glycerol and 0.5% Tween 80) was added. To the first rows of the plate, 100 µL of the pure compound, nanoparticle formulations as well as anti-tuberculosis drugs at the appropriate concentrations were added in duplicate as the positive control. The pure compound, 7-MJ, was prepared by dissolving 7-MJ in 10% DMSO in sterile 7H9 Middlebrook media, resulting in a stock solution of 1 mg/mL. A two-fold serial dilution of media containing 7-MJ in wells in row one was carried out to obtain a concentration range from 100 µg/mL to 0.04 µg/mL in a volume of 100 µL per well. Two-fold serial dilutions of the blank nanoparticles and the 7-MJ nanoparticles resulted in a concentration range from 1000 µg/mL to 0.49 µg/mL to. The positive drug controls, rifampicin, isoniazid, and streptomycin, were prepared to final concentrations to provide a range from 3.2 µg/mL to 0.002 µg/mL, 6.4 µg/mL to 0.004 µg/mL, and 100 µg/mL to 0.04 µg/mL, respectively, during the 2-fold serial dilution across the 96-well plate. An equivalent amount of DMSO used to dissolve the 7-MJ and the anti-tuberculosis drugs and wells containing cells only (uninfected) were included as positive and uninfected controls. Both *M. tuberculosis* strains were grown in Middlebrook 7H9 broth supplemented with 0.2% glycerol, 10% Middlebrook oleic acid-albumin-dextrose-catalase (OADC), and 0.05% Tween 80 at 37 °C for 3–4 days to an OD_600nm_ = 0.5–0.8. The bacterial cultures were diluted 1:500 in 7H9 medium and used as the inoculum to determine the MIC of each of the compounds required to inhibit growth of the *M. tuberculosis* strains. The plates were incubated for a total of 14 days in a CO_2_ incubator at 37 °C. The plates were scored for growth as pellets at the bottom of the well at 7 days and 14 days using an inverted mirror. The last row where no growth was observed represented the MIC for the compound. To conform to the MIC, Alamar Blue was added to each of the well on day 14, and the plates incubated overnight before visually scoring for a change in color from pink (viable growth) to blue (no growth) using an inverted mirror.

### 2.5. In Vitro Cytotoxicity Activity—U937 Cell Line

The method to determine the degree of cytotoxicity exhibited by the pure compound and all the respective PLGA nanoparticles (blank nanoparticles, 7-MJ nanoparticles, and 7-MJ + RhB nanoparticles) was conducted according to Lall et al. (2016) [16]. The human histiocytic lymphoma cell line (U937) was utilized in this assay and cultured in RPMI 1640 media supplemented with fetal bovine serum (FBS) and antibiotics until 90% confluency was achieved. The tissue culture was maintained at 37 °C and 5% CO_2_ in a CO_2_ incubator. Once adequate confluency had been achieved, 0.1% Phorbol 12-myristate 13-acetate (PMA) was added at 1 µg/mL to stimulate the lymphoma cell line to differentiate into macrophage cells. Subsequently, 100 µL of the stimulated cells were plated in a sterile flat-bottom 96-well tissue plate at a concentration of 100,000 cells/mL. The controls in this experiment included a RPMI 1640 media-only control, a 1% DMSO solvent control for 7-MJ, a 1% dH_2_O solvent control for the nanoparticles, and an untreated cell control. The positive controls utilized were Actinomycin D and DMSO at a range of 0.002 to 0.05 μg/mL and percentage range of 0.625 to 20%, respectively. The nanoparticles and pure compound were tested at final concentrations which ranged from 12.5 to 400 μg/mL and 0.156 to 5 μg/mL, respectively. For RhB, 1 mg of RhB was dissolved in 1 mL dH_2_O and tested at concentrations ranging from 0.625 to 20 μg/mL. In a CO_2_ incubator, the plate containing the cells was incubated for 72 h at 37 °C. Following the incubation, 20 µL of PrestoBlue was added to all the wells and further incubated for 3 h at 37 °C in a CO_2_ incubator. A Victor Nivo (PerkinElmer, Midrand, South Africa) microplate reader with excitation/emission values of 560/595 nm was used to evaluate the absorbance. The half-maximal inhibitory concentration (IC_50_) values for the samples and the controls were determined using GraphPad Prism 4™ (Version 4).

### 2.6. Engulfment Studies

To determine the quantity of PLGA nanoparticles engulfed by differentiated U937 macrophage-like cells, cells were cultured in RPMI 1640 media supplemented with FBS and antibiotics to achieve 90% confluency. The tissue culture was maintained at 37 °C and 5% CO_2_. Once adequate confluency had been achieved, the cells were seeded at 100,000 cells/mL in T25 flasks, and PMA was added at 1 μg/mL. After 24 h of incubation, the cells were treated with 7 MJ + RhB nanoparticles at various concentrations (50 μg/mL, 100 μg/mL, 125 μg/mL, and 150 μg/mL). The controls included untreated cells containing media and positive controls of 10 µL and 15 µL of RhB (0.1% *w*/*v*). Following 72 h incubation at 37 °C, the cells were washed with PBS. Trypsin was added followed by additional incubation of cells for 1 min. A cell count of each sample was conducted using a Countess II FL Automated Cell Counter (Thermo Fisher Scientific, Waltham, MA, USA), where 10,000 of the live counted cells were added to 1 mL of RPMI 1640 media. The cells were then run through a BD Accuri™ C6 Plus Flow Cytometer (BD Biosciences, Franklin Lakes, NJ, USA) preconfigured with a green (533/30) standard optical filter and using the FL2-A sensors. The number of events recorded was set to 5000, and the flow rate was set to slow. The data were quantified using FlowJo™ (Version 10.8.1.8) software, the cell populations were gated, and histograms were produced of counted cells versus fluorescence intensity.

### 2.7. Statistical Analysis

The data reported in this study were acquired in triplicate, the mean of three independent repeats of each experiment was calculated and is shown as the mean ± standard deviation (SD). For the IC_50_ values analysis, GraphPad Prism 4™ software (Version 4) was used. Using a 4-parameter sigmoidal dose–response curve, the absolute IC_50_ values were calculated. Constraints were set on the top (100) and bottom parameters (0). The statistical analysis of the quantified flow cytometry data was performed using RStudio software (Version 2022.07). As a post hoc analysis, a one-way analysis of variance (ANOVA) was completed together with Tukey’s multiple comparisons tests. A *p*-value less than 0.05 was regarded as statistically significant when comparing the nanoparticle treatment to the positive control.

## 3. Results

### 3.1. Dynamic Light Scattering Analysis

For the determination of the hydrodynamic diameter, polydispersity index (PdI), and zeta potential of the formulated nanoparticles, the dynamic light scattering technique was utilized, with results depicted in Table 1. Hydrodynamic diameter is an indication of the size of the nanoparticles, and PdI is a measurement of the homogeneity of the nanoparticles. The zeta potential is the potential difference between the dispersion medium and the stationary layer of fluid attached to the dispersed particles also known as the electrokinetic potential of the nanoparticles in colloidal dispersions. Therefore, the zeta potential indicates the surface charge and potential stability of the nanoparticles in suspension. The mean hydrodynamic diameter of RhB was shown to be 1763.67 ± 53.15 nm (PdI = 0.66 ± 0.12).

### 3.2. Drug Analysis

The ultra-violet (UV-Vis) spectra of the 7-MJ nanoparticles and the 7 MJ + RhB nanoparticles was used to determine the estimation of the drug loading capacity and entrapment efficiency percentages, as shown in Table 2. The pure compound, 7-MJ, was observed at a wavelength of 430 nm and absorbance values between 0.22 and 0.78 A. The fluorescent dye, RhB, was observed at a wavelength and absorbance of 554.40 nm and 3.61 A, respectively.

The entrapment efficiency of the pure compound, 7-MJ, in the 7-MJ and 7 MJ + RhB nanoparticles was 50.43% and 61.85%, respectively. The drug loading content of the 7-MJ in the 7-MJ and 7 MJ + RhB—nanoparticles was 5.73% and 6.72%, respectively. The entrapment efficiency and drug loading content of RhB in the 7 MJ + RhB nanoparticle was 83.79% and 25.39%, respectively.

### 3.3. Fourier Transform Infrared Spectroscopy

Fourier transform infrared spectroscopy was conducted on all the dried synthesized PLGA nanoparticles (Figure 1).

The wide and sharp band between 3000 and 3500 cm^−1^ on the FTIR spectrum of the PLGA nanoparticles is due to the -OH group, which appears at one end of the PLGA copolymers when formulated into PLGA nanoparticles [25,26]. The intense band observed at around 1750 cm^−1^ is attributed to the stretching vibration of the carbonyl groups present in the two monomers [27]. The sharp peak at about 1000 cm^−1^ is attributed to C-O stretching of the ester group present in PLGA and the PLGA nanoparticles [28]. Stretching vibrations of the C-H group can be observed at around 3000 cm^−1^, whereas bending vibrations of the C-H group were observed at the peak around 1250 cm^−1^. In the FTIR spectrums of the 7-MJ nanoparticles and the 7 MJ + RhB nanoparticles, peaks were observed at around 1580–1650 cm^−1^. These peaks could be due to the bending N-H groups of the 7-MJ compound [29]. The peaks found at 1200 and 1450 cm^−1^ in the FTIR spectrum of the 7 MJ + RhB nanoparticles seem to differ compared to the blank and 7-MJ nanoparticles. This could be due to the C-N vibration and the NH_2_ bending vibrations of RhB found together with the 7 MJ + RhB nanoparticles.

### 3.4. X-Ray Diffraction Analysis

To confirm the crystalline structure of the PLGA nanoparticles, X-ray diffraction (XRD) analysis was used (Figure 2).

The XRD patterns of the blank, 7-MJ, and 7 MJ + RhB nanoparticles (Figure 2A) showed low intensity peaks across the range of 10 to 30 θ, forming a dome-shaped region. The 7-MJ + RhB nanoparticles showed another small dome-shaped region with low intensity peaks across the range of 40 to 50 θ. PLGA by itself (Figure 2B) also showed low intensity peaks at around 20 θ, forming a small dome. On the other hand, pure 7-MJ and RhB (Figure 2B) showed more intense peaks throughout the scan.

### 3.5. In Vitro Antimycobacterial Activity Against Mycobacterium smegmatis and Mycobacterium tuberculosis

The in vitro antimycobacterial activity of the blank nanoparticles, the 7-MJ nanoparticles formulations, and the pure compound, 7-MJ, was assessed qualitatively using the broth micro-dilution method to determine the minimum inhibitory concentration (MIC) values as shown in Table 3.

The in vitro broth microdilution assay showed that 7-MJ had an MIC value of 1.60 µg/mL, compared to 125 µg/mL for 7-MJ nanoparticles against *M. smegmatis*. 7-MJ also displayed in vitro antimycobacterial activity against drug-susceptible *M. tuberculosis* (H37Rv) and multi-drug-resistant *M. tuberculosis* (MDR11) with MIC values of 0.4 µg/mL and 1.60 µg/mL, respectively. The 7-MJ nanoparticles showed MIC values of 250 µg/mL for both the drug-susceptible and multi-drug-resistant *M. tuberculosis* strains. As expected, the blank nanoparticles had MIC values of >1000 µg/mL for all mycobacterial strains tested. Therefore, the activity shown by the 7-MJ nanoparticles was due to 7-MJ and not due to the PLGA nanoformulation components.

### 3.6. In Vitro Cytotoxicity

The pure compound (7-MJ), the fluorescent dye (RhB), and the various PLGA nanoformulations (blank nanoparticles 7-MJ nanoparticles and 7 MJ + RhB nanoparticles) were investigated for their cytotoxicity in U937 cells, and their IC50 values are presented in Table 4.

To determine the ethical viability of a drug for use as a therapeutic intervention in the treatment of disease, it is necessary to conduct studies which determine the degree of toxicity exhibited by the drug, as presented in the dose–response curves in Figure 3.

According to the guidelines of the National Cancer Institute (NCI), a compound is classified as displaying high cytotoxicity at IC50 values ≤ 20 µg/mL; moderate cytotoxicity at IC50 values between 20 and 200 µg/mL, weak cytotoxicity at IC50 values between 200 and 500 µg/mL, and no cytotoxicity at IC50 values > 500 µg/mL [30,31]. The IC50 of 7-MJ, blank, 7-MJ, and 7 MJ + RhB nanoparticles were determined on differentiated U937 cells, as seen in Table 4. The IC50 value of 7-MJ was 3.25 ± 0.22 µg/mL, >400 µg/mL for the blank nanoparticles, 247.17 ± 3.79 µg/mL for the 7-MJ nanoparticles, and 104.33 ± 4.96 µg/mL for the 7 MJ + RhB nanoparticles. The blank nanoparticles showed weak cytotoxicity on the differentiated U937 cells; therefore, any cytotoxicity exhibited by the 7-MJ nanoparticles and the 7 MJ + RhB nanoparticles may have potentially been due to the entrapped compound. According to previous studies conducted as well as verified in the current study, 7-MJ was classified as highly cytotoxic.

### 3.7. Engulfment Studies

The cellular uptake of 7-MJ + RhB nanoparticles into U937 differentiated macrophages was examined using flow cytometry. The histograms in Figure 4 illustrate a shift to the right resulting from the detection of fluorescence of RhB in treated cell populations compared to the unstained control cell population. The M1 section of the diagram represents the autofluorescence of the non-treated cells with the additional intensity of fluorescence caused by the uptake of RhB, demonstrated as the M2 section of the diagram.

The proportion of cells in the M2 section of the flow cytometric histograms that indicated the detection of fluorescence events is represented in Figure 5. The cells treated with the 7 MJ + RhB nanoparticles exhibited a significantly lower mean fluorescence intensity (MFI) compared to the RhB positive controls.

## 4. Discussion

Characterization of nanoparticles is required and important as it assesses the physical and chemical properties and quality of the nanoformulations [32,33]. The difference in size between the blank nanoparticles (257.10 ± 2.97 nm) and the 7-MJ nanoparticles (288.27 ± 4.76 nm) could possibly be attributed to the successful entrapment of 7-MJ in the 7-MJ nanoparticles. Nanoparticles that have a mean hydrodynamic diameter in a range of 100–300 nm exhibit favorable cellular uptake, as they can efficiently enter cells via endocytic mechanisms and have a higher chance of accumulating at target sites [34,35]. Nanoparticles in the range of 100–500 nm often exhibit better stability as there is a balance between Brownian motion, which tends to disperse particles, and aggregation forces, which bring particles together [36]. The mean hydrodynamic diameter of 7 MJ + RhB nanoparticles (667.30 ± 15.70 nm) is much larger than the desirable size of ±300 nm. Particles with mean hydrodynamic diameters larger than 500 nm are internalized into cells through macropinocytosis and/or phagocytosis. However, these mechanisms have lower uptake efficiency compared to other endocytic processes. For example, there is micropinocytosis, which can only take up particles with a maximum mean hydrodynamic diameter of 200–300 nm. [35,37]. The large size of the 7 MJ + RhB nanoparticles could be due to the large hydrodynamic diameter of the fluorescent dye, RhB, of 1763.67 ± 53.15 nm.

Polydispersity index measurements vary from 0 to 1, where 0 refers to a perfectly uniform sample and 1 refers to a highly polydisperse sample [38]. The observed PdIs of the blank nanoparticles (0.10 ± 0.009) and the 7-MJ nanoparticles (0.09 ± 0.03), are more uniform in size and displayed better homogeneity than the 7 MJ + RhB nanoparticles (0.59 ± 0.03) (Table 1). The polydispersity of the 7 MJ + RhB nanoparticles can also be attributed to the fluorescent dye, RhB, as it had a PdI value of 0.66 ± 0.12. This indicates that the dye not only affected the size of the 7 MJ + RhB nanoparticles but also the homogeneity of the particles.

PVA was added during the synthesis of the PLGA nanoformulations to enhance the stability thereof. A PVA coating imparts steric stabilization which helps to prevent the aggregation of the nanoparticles [27]. The larger the zeta potential value of particles in a suspension, which is the case for the 7 MJ + RhB nanoparticles (−20.50 ± 0.45 mV as seen in Table 1), the more likely the stability. This is due to the charged particles repelling each other and overcoming the natural tendency to aggregate [39]. A zeta potential beyond ± 30 mV is most favorable, as it provides sufficient electrostatic stabilization [40,41]. The zeta potential of around −14 mV exhibited by the blank nanoparticles and 7-MJ nanoparticles can still contribute to the stability of the nanoparticle suspension as it still provides repulsive forces between particles of the same charge and prevents agglomeration to some extent [42].

In order to determine the EE and DLC capabilities of the PLGA nanoparticles, UV-Vis spectroscopy was conducted, as shown in Table 2. This analysis utilizes visible light to determine the concentration of chemical compounds in a solution [43]. EE refers to the percentage of the drug loaded relative to the total amount of drug. Whereas DLC refers to the percentage of the drug loaded relative to the total mass of the nanoparticle [44]. There is a clear difference in the EE and DLC percentages of 7-MJ and RhB. 7-MJ and PLGA are both hydrophobic, whereas RhB is hydrophilic. A possible reason for the higher percentages of RhB could be that the nanoformulations containing RhB were prepared using a W/O/W emulsion. A W/O/W emulsion consists of internal aqueous phase droplets within larger oil droplets, which are dispersed in an external aqueous phase. In this structure, RhB is contained in the inner aqueous droplets, while 7-MJ and PLGA are in the oil phase (Appendix A). Since RhB has low affinity for the oil phase, it remains entrapped in the core of the PLGA nanoparticle matrix [45,46]. RhB’s retention in the core of the nanoparticle matrix helps reduce the initial burst release [47]. On the other hand, 7-MJ is incorporated more towards the surface of the nanoparticles. Due to being close to the surface and the lack of strong intermolecular interactions between 7-MJ and PLGA, 7-MJ is more susceptible to initial burst release or leakage [48,49]. This can explain the differences in EE and DLC percentages of 7-MJ and RhB within the nanoformulations.

To evaluate the physicochemical state and possible interactions between the PLGA nanoparticles and the incorporated compounds, FTIR and XRD were used. Similar peaks were observed in FTIR analysis of different studies incorporating PLGA polymers, thus indicating the successful synthesis of PLGA nanoparticles in the present study [27]. For slow release to be achieved, the compounds need to bind to the matrix of the nanoparticle, which is the case in this study for both the pure compound, 7-MJ, and fluorescent dye, RhB, as they produced their own peaks, as discussed in the results and indicated in the FTIR spectra in Figure 1. This indicates that the entrapped compounds will most likely show slow-release patterns, as Figure 1B confirms the successful incorporation of 7-MJ and RhB into the respective nanoparticles. Further release studies will, however, have to be conducted to confirm this observation.

The XRD patterns in Figure 2A of the blank, 7-MJ, and 7 MJ + RhB nanoparticles exhibited very small areas under the peaks. Therefore, it can be concluded that the nanoformulations were mostly amorphous rather than crystalline in nature [45,50]. Figure 2B exhibits that the pure 7-MJ and RhB did not affect the amorphous nature of PLGA, which remained amorphous after nanoparticle synthesis. An amorphous solid is defined as a non-crystalline solid because its molecules and atoms are not organized in a definite order, therefore lacking a definite form [51,52]. An advantage of an amorphous nanoparticle, compared to a crystalline nanoparticle, is higher kinetic solubility [53]. A higher kinetic solubility means that the nanoparticles will have higher solubility and dissolution rates, which in turn can result in better absorption and bioavailability of the nanoparticles within the body [54].

The formulated nanoparticles remained stable in storage conditions of −20 °C as the mean hydrodynamic diameter, PdI, zeta potential, and pH of the nanoparticles over the period of 6 months did not change significantly as summarized in Appendix A). A pH of around ±8 was exhibited by all three of the nanoparticles, indicating a near-neutral solution (pH 6–8) [55]. This further indicated that the PLGA nanoparticles stayed stable in their storage conditions, as rapid degradation of PLGA into its monomers, lactic acid and glycolic acid, produce local transient acidification [56]. The in vitro stability analysis conducted over a period of 6 days in various biological mediums is depicted in Appendix A); the nanoparticles exhibited stable mean hydrodynamic diameters for the duration of the experiments.

In Table 3, the antimycobacterial activity of 7-MJ and the nanoparticles is detailed. The MIC values exhibited by 7-MJ against *M. smegmatis* (1.60 µg/mL) and drug-susceptible *M. tuberculosis* (0.4 µg/mL) are comparable to previously reported values [17,18]. In this study, it was indicated that 7-MJ also exhibits promising antimycobacterial activity against multi-drug-resistant *M. tuberculosis* (MDR11) with an MIC value of 1.60 µg/mL. The MIC values exhibited by the 7-MJ nanoparticles could be due to the slow release of 7-MJ, which, as previously shown, is known to be active against both *M. smegmatis* and *M. tuberculosis* (H37Rv).

The high cytotoxic effect of 7-MJ on U937 cells was also reported in a previous study by Kishore et al. (2014), with an IC_50_ of between 1 and 5 µg/mL, which is comparable to the IC_50_ of 3.25 ± 0.22 µg/mL reported in the current study [15]. In other studies, the cytotoxicity of 7-MJ was tested on the mouse macrophage (J774A.1) cell line, green monkey kidney (Vero) cell line, and peripheral blood mononuclear cells (PBMCs), exhibiting IC_50_ values of 3.91 µg/mL, 15.11 µg/mL, and 3.46 µg/mL, respectively [15,16]. This further supports the findings of the highly toxic nature of 7-MJ. The 7-MJ nanoparticles were found to be weakly cytotoxic. As indicated by the current study, the entrapment of 7-MJ within the PLGA nanoparticles reduced the cytotoxic effect of 7-MJ by approximately 80-fold.

The macrophage-like U937 cell line is commonly used in nanoparticle uptake studies due to its ability to internalize small carriers for inflammation or infection treatment [57,58,59]. PLGA has been shown to be taken up by macrophages, the primary cells targeted during *M. tuberculosis* infection [60,61,62]. Fenaroli et al. demonstrated co-localization of PLGA nanoparticles and mycobacteria in macrophages using a zebrafish model [63]. To assess PLGA nanoparticle uptake by U937 macrophages, PLGA nanoparticles containing 7-MJ were fluorescently labelled with RhB, allowing for visual tracking and quantification via flow cytometry [64]. In Figure 4, the histograms were divided into M1 (autofluorescence from untreated cells) and M2 (RhB fluorescence) regions [65]. A rightward shift in cells treated with 7-MJ + RhB nanoparticles (Figure 4A–D) indicated nanoparticle uptake, while cells treated with free RhB showed a more prominent shift (Figure 4E,F), suggesting a higher uptake of RhB. Differences in fluorescence intensity between free RhB and RhB-loaded PLGA nanoparticles may result from factors like synthesis conditions, dye–polymer interactions, nanoparticle size, or incomplete uptake of the 7-MJ + RhB nanoparticles [66,67,68].

The decreased fluorescence intensity of the 7 MJ + RhB nanoparticles is unlikely due to low entrapment efficiency, as around 83.54% of the 0.02 µg/mL RhB was encapsulated, resulting in approximately 0.016 µg/mL of RhB in the formulation (Table 2). The free RhB tested at concentrations of 0.01 and 0.015 µg/mL was consequently expected to show similar fluorescence intensity to the RhB in the nanoparticles. To investigate if the reduced fluorescence was due to partial nanoparticle uptake, cells were treated with varying concentrations of 7 MJ + RhB nanoparticles (50, 100, 125, and 150 µg/mL), and the fluorescence was quantified (Figure 5). If incomplete uptake were the cause, higher nanoparticle concentrations should increase fluorescence intensity [69,70]. However, the percentage of cells showing fluorescence uptake remained similar (54.1–61.9%) across concentrations, suggesting that this could also not have been the cause. The IC_50_ of the 7 MJ + RhB nanoparticles (104.33 ± 4.96 µg/mL) was also taken into consideration, and, therefore, the cells were seeded at a much higher concentration (100,000 cells/mL). The possible reason for the lower fluorescence intensity of RhB in the 7 MJ + RhB nanoparticles could be due to RhB being confined within the PLGA matrix, restricting its movement compared to free RhB in solution [71,72,73,74]. This limitation may reduce RhB’s exposure to excitation light, thus lowering the fluorescence exhibited by the 7 MJ + RhB nanoparticles. Additionally, the nanoparticle size could contribute, as PLGA nanoparticles larger than 500 nm may be less efficiently internalized by macrophages [35,37].

This study highlights the importance of factors such as size, charge, pH, stability, and physicochemical properties when formulating PLGA nanoparticles for drug delivery. Different nanoparticle formulations can impact efficacy and application [75,76,77]. Before delivering drugs to target cells, it is essential to assess whether nanoparticles can be taken up by cells and their potential cytotoxicity. In this study, we evaluated the uptake of 7 MJ + RhB PLGA nanoparticles by U937 macrophage-like cells, the primary target for TB mycobacteria. Although the nanoparticles were larger than the optimal size (<300 nm), they were still successfully engulfed by the cells. Optimizing nanoparticle size could enhance uptake and stability. Previous studies, including this current study, suggest that PLGA nanoformulations may reduce the cytotoxicity of toxic compounds in vitro. However, no FDA-approved PLGA-based drug formulations are currently on the market, although nineteen FDA-approved PLGA products exist, mainly in the form of in situ gels, implants, and microparticles [78]. This underscores the need for further efficacy and safety evaluations of PLGA nanoformulations before clinical approval. Despite these challenges, ongoing research into biocompatibility, cytotoxicity reduction, and FDA approval pathways for PLGA formulations continues to show promise.

## 5. Conclusions

Important standards to keep in mind regarding the toxicity and efficacy of PLGA nanoparticles are physiochemical properties such as surface charge and particle size. This study confirmed that 7-MJ as well as the 7-MJ nanoparticles had activity against *M. smegmatis*, drug-susceptible *M. tuberculosis* (H37Rv), and multi-drug-resistant *M. tuberculosis* (MDR11). In support of the hypothesis, the PLGA nanoparticles reduced the cytotoxicity of 7-MJ considerably, and U937 macrophage cells were able to successfully engulf the 7 MJ + RhB nanoparticles. Future studies should examine the cytotoxic effect of the PLGA nanoparticles on other cells of interest, such as cells involved en route to the target site, for example, the human alveolar macrophage-like (Daisy) cell line, human lung fibroblast (MRC-5) cell line, and human hepatoma (HepG_2_) cell line. Additionally, drug release studies, testing the activity of the 7-MJ PLGA nanoformulation against other drug-resistant strains of *M. tuberculosis*, biofilm inhibition, and the enzyme inhibition of mycothiol disulfide reductase can be considered. These additional studies will enable the evaluation of the amount of 7-MJ that is released and for how long 7-MJ will be slowly released from PLGA nanoparticles as well as broaden the potential use of this promising drug.

## Figures and Tables

**Figure 1 pharmaceutics-16-01477-f001:**
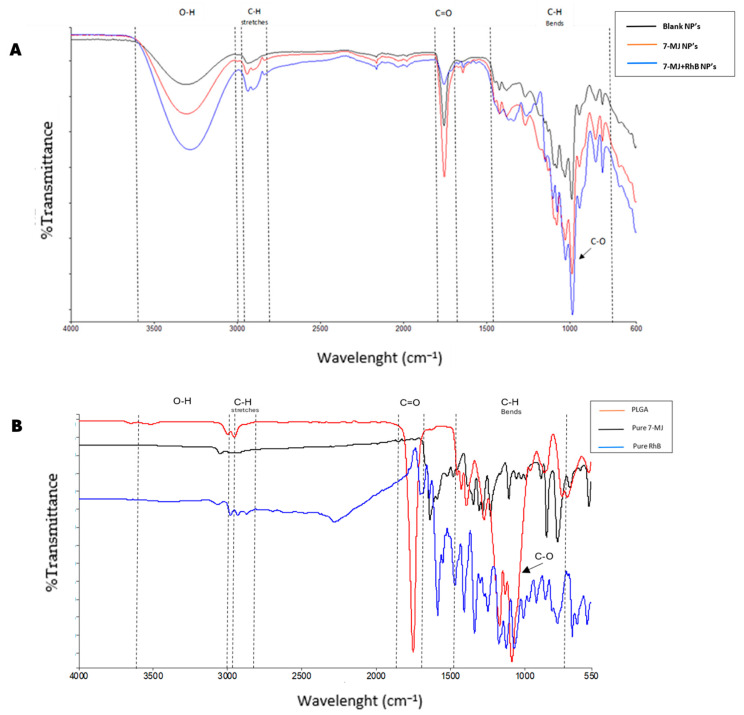
FTIR spectra of (**A**) the blank nanoparticles, 7-MJ nanoparticles, and 7 MJ + RhB nanoparticles and (**B**) PLGA, pure 7-MJ, and RhB.

**Figure 2 pharmaceutics-16-01477-f002:**
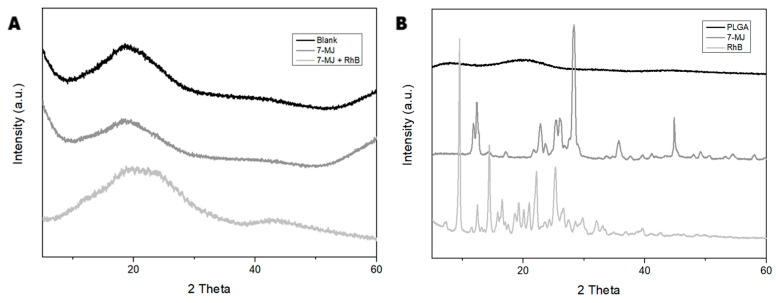
XRD analysis graphs of (**A**) the PLGA nanoformulations and (**B**) the pure compounds.

**Figure 3 pharmaceutics-16-01477-f003:**
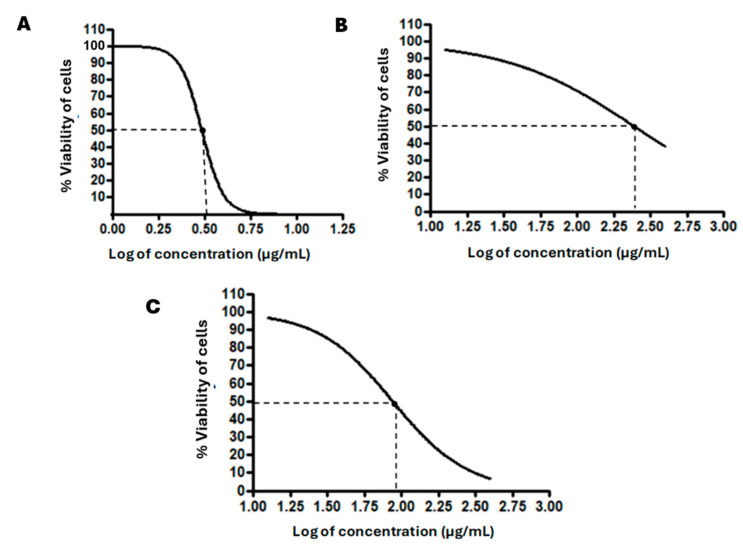
Dose–response curves representing the percentage viability of the cells after treatment with (**A**) 7-MJ, (**B**) 7-MJ nanoparticles, and (**C**) 7-MJ + RhB nanoparticles. The dotted lines on the curves represent the log of the IC50 values of each respective compound.

**Figure 4 pharmaceutics-16-01477-f004:**
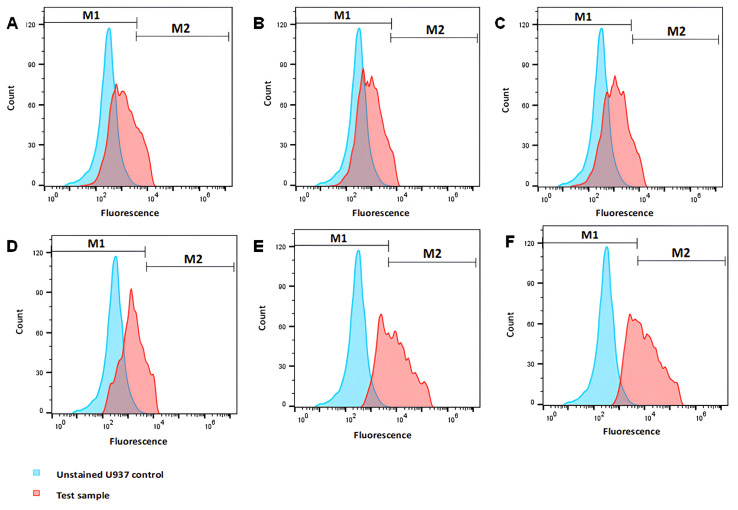
Flow cytometry histograms showing cellular uptake of 7 MJ + RhB nanoparticles into U937 differentiated macrophages. The blue graph represents the unstained control cell population, and the red graph represents cells treated with the 7 MJ + RhB nanoparticle at concentrations of (**A**) 50 μg/mL, (**B**) 100 μg/mL, (**C**) 125 μg/mL, and (**D**) 150 μg/mL, including the positive controls of free RhB (0.1%) at (**E**) 10 µL and (**F**) 15 µL of RhB. M1 indicates autofluorescence of the non-treated cells, and M2 indicates the intensity of fluorescence caused by RhB.

**Figure 5 pharmaceutics-16-01477-f005:**
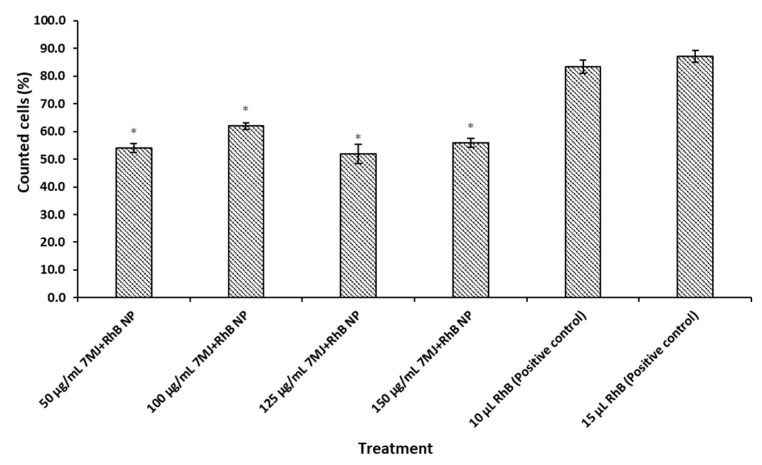
Percentage of cells counted as fluorescence events following flow-cytometry analysis after treatment of cells with different concentrations (50 μg/mL, 100 μg/mL, 125 μg/mL, and 150 μg/mL) of 7 MJ + RhB nanoparticles as well as two different concentrations of the RhB positive control (0.01 μg/mL and 0.015 μg/mL). * *p* < 0.05.

**Table 1 pharmaceutics-16-01477-t001:** The DLS analysis conducted on the PLGA nanoformulations.

Nanoparticle	Mean Hydrodynamic Diameter (nm)	PdI ^1^	Zeta (ζ) Potential (mV)
Blank nanoparticles	257.10 ± 2.97	0.10 ± 0.009	−14.27 ± 0.76
7-MJ nanoparticles	288.27 ± 4.76	0.09 ± 0.03	−14.00 ± 0.15
7-MJ + RhB nanoparticles	667.30 ± 15.70	0.59 ± 0.03	−20.5 ± 0.45

^1^ Polydispersity index.

**Table 2 pharmaceutics-16-01477-t002:** The UV-Vis spectroscopy, drug loading capacity (%), and entrapment efficiency (%) of the PLGA nanoparticle formulations.

Nanoparticle	DLC ^1^ (%) of 7-MJ	EE ^2^ (%) of 7-MJ	DLC ^1^ (%) of RhB	EE ^2^ (%) of 7-RhB
7-MJ nanoparticles	5.39 ± 0.23	50.99 ± 3.55	n/a *	n/a *
7-MJ + RhB nanoparticles	6.85 ± 0.80	67.29 ± 4.73	25.31 ± 1.68	83.54 ± 5.54

^1^ Drug loading content, ^2^ Entrapment efficiency, * n/a: not applicable.

**Table 3 pharmaceutics-16-01477-t003:** The UV-Vis spectroscopy, drug loading capacity (%), and entrapment efficiency (%) of the PLGA nanoparticle formulations.

	*Mycobacterium smegmatis*	*Mycobacterium tuberculosis* (H37Rv)	*Mycobacterium tuberculosis* (MDR11)
Samples	MIC ^1^ (μg/mL)	MIC ^1^ (μg/mL)	MIC ^1^ (μg/mL)
Pure compound			
7-Methyljuglone (7-MJ)	1.60	0.40	1.60
PLGA nanoparticle formulations			
Blank nanoparticles	>1000	>1000	>1000
7-MJ nanoparticles	125	250	250
Controls			
Ciprofloxacin	0.31	n/a *	n/a *
Isoniazid	n/a *	0.10	>6.4
Streptomycin	n/a *	0.40	6.30
Rifampicin	n/a *	0.002	>3.20

^1^ Minimum inhibitory concentration, * n/a: not applicable.

**Table 4 pharmaceutics-16-01477-t004:** The effect of the pure compound, fluorescent dye, and PLGA nanoformulations on the cell proliferation of U937 cells after 72 h treatment.

Samples	IC_50_ ^1^ ± SD ^2^(μg/mL)
Pure compound and fluorescent dye	
7-Methyljuglone (7-MJ)	3.25 ± 0.22
Rhodamine B (RhB)	>20
PLGA nanoparticle formulations	
Blank nanoparticles	>400
7-MJ nanoparticles	247.17 ± 3.79
7-MJ + RhB nanoparticles	104.33 ± 4.96
Controls	
Actinomycin D	0.04 ± 0.01
DMSO 20%	6.60 ± 1.14

^1^ Inhibitory concentration whereby 50% of the bacterial growth was inhibited, ^2^ Standard deviation.

## Data Availability

Data is contained within the article and Appendix A.

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
