# Peer review of "Antitubercular Activity of 7-Methyljuglone-Loaded Poly-(Lactide Co-Glycolide) Nanoparticles"

_pharmaceutics, 2024, doi:10.3390/pharmaceutics16111477_

Round 1
Reviewer 1 Report
Comments and Suggestions for Authors
In the current research article entitled Antitubercular activity of 7-methyljuglone-loaded poly- (lactide co-glycolide) nanoparticles, authors provide a comprehensive study about preparation and charaterization of poly- (lactide co-glycolide) nanoparticles loaded with 7-methyljuglone. However, several issues need to be improved throughout the manuscript, which contains a lot of repeated information that increases the length of the manuscript without any demand. Please find below some demands for improving the manuscript.
Line 25: Please ensure using abbreviations throughout the manuscript. (7-Methyljuglone → 7-MJ).
Line (56-60) (69-73): Please avoid using long sentences. Try to make it simple for readers.
Line 82: Please define the abbreviation for the first time it appears in the manuscript (7-MJ).
Line 106: Please define the abbreviation (RhB)
Line 109: The whole synthesis process is not written very well.
What is the oil phase?
Why mention (50:50) in the following context → 50 mg of PLGA 50:50?
Please explain the above issues and make a simple sketch to show all the procedures for nanoparticle preparation.
Line 156: Please rename this section to a method for drug analysis. Please make it directly below Chemicals, reagents, and pure compound.
Line 159: drug content is usually measured at a fixed wavelength. How drug content was measured between 190 – 800 nm?
Line 162: Please change number for this section to be
2.2.4. Drug loading content (DLC) and Encapsulation efficiency (EE) of the PLGA nanoparticles
Line 313 – 336: Please try to avoid repetition. Most of the information in the first paragraph is similar to the second paragraph and table. Just give a brief paragraph about the table.
Line 361: if applicable, please make FTIR scan for pure 7-MJ and RhB.
Line 381: if applicable, please make XRD scan for pure 7-MJ and RhB.
Line 450: The discussion part is too long and contains some simple principles. Please try to make it simple and attractive to readers.
Comments on the Quality of English Language
Some sentences were hard to understand.
Also, use sentences that are shorter.
Author Response
We would like to express our sincere gratitude for the time and effort you dedicated to reviewing our manuscript. Your insightful comments and constructive suggestions have been incredibly helpful in improving the quality and clarity of our work. We truly appreciate your thoughtful consideration and the valuable feedback you provided during the review process.
Please see below our response to your comments:
Comment 1: Line 25: Please ensure using abbreviations throughout the manuscript. (7-Methyljuglone → 7-MJ).
Response 1: Thank you for your helpful comment. We apologize for any inconsistencies in the use of abbreviations. We have now ensured that the abbreviation 7-MJ is used consistently throughout the manuscript, including in the abstract, introduction, and results sections, as suggested. We appreciate your attention to detail, which has helped improve the clarity and consistency of the manuscript.
Comment 2: Line (56-60) (69-73): Please avoid using long sentences. Try to make it simple for readers.
Response 2: Thank you for your valuable feedback. We appreciate your suggestion to simplify the sentence structure for better readability. In response, we have revised the manuscript to reduce the length of the above-mentioned sentences and ensure that the text is clearer and more concise for the readers.
For Line (56-60) the long sentence of “Some of the major advantages of using nanoparticles as drug delivery systems are their ability to improve the aqueous solubility of poorly soluble drugs, incorporate both hydrophobic and hydrophilic drugs, protect the drugs from degradation and allow controlled release of the drug, thereby reducing the frequency of administration and dose” was broken down and simplified to “Nanoparticles offer several key advantages as drug delivery systems. They can improve the aqueous solubility of poorly soluble drugs. Nanoparticles can also carry both hydrophobic and hydrophilic drugs. Additionally, they protect drugs from degradation. Finally, nanoparticles allow controlled release, which can reduce the frequency of administration and the required dose.”
For line (69-73) the long sentence of “Poly (lactic-co-glycolic acid) nanoparticles have been previously tested for improved TB treatment as it offers several advantages, one of which is their ability to accumulate in Mycobacterium tuberculosis (M. tuberculosis) -infected macrophages, which are the primary immune host cells that control the infection.” was shortened and simplified to “PLGA nanoparticles have been studied for improving TB treatment. One key advantage is their ability to accumulate in Mycobacterium tuberculosis (M. tuberculosis)-infected macrophages. These macrophages are the primary immune cells responsible for controlling the infection”
We hope these revisions improve the overall flow and readability of the manuscript.
Comment 3 and 4: Line 82 and 106: Please define the abbreviation for the first time it appears in the manuscript (7-MJ and RhB).
Response 3 and 4: Thank you for your helpful comment, which improved the clarity and precision of the manuscript. We have now ensured that both 7-MJ and RhB are defined the first time they appear in the manuscript.
Comment 5: Line 109: The whole synthesis process is not written very well.
What is the oil phase?
Why mention (50:50) in the following context → 50 mg of PLGA 50:50?
Please explain the above issues and make a simple sketch to show all the procedures for nanoparticle preparation.
Response 5: Thank you for your feedback. We apologize for any lack of clarity in the description of the synthesis process. We have revised this section to provide better clarity. Please see below the clarifications to your questions and the incorporated revisions. We hope these revisions improve the clarity and comprehensibility of the synthesis process. Thank you again for your valuable feedback.
For your question regarding the oil phases. The oil phase of the blank nanoparticles consisted of PLGA dissolved in acetone. Whereas the oil phase of the 7-MJ nanoparticles and 7-MJ + RhB nanoparticles consisted of PLGA and 7-MJ dissolved in acetone
For your question regarding the mention of PLGA 50:50. The 50:50 refers to the ratio of lactic acid and glycolic acid in the PLGA co-polymer, so the PLGA consisted of 50% lactic acid and 50% glycolic acid. This is mentioned as there are various different ratios in which the monomers can occur to form PLGA.
We have revised the section on PLGA nanoparticle synthesis and rewrote various paragraphs for better clarity on both concerns:
The following was changed “Three PLGA nanoformulations were prepared; blank nanoparticles, 7-MJ nanoparticles, and 7-MJ+RhB nanoparticles. The PLGA nanoparticles were prepared through a nanoprecipitation technique. For the blank nanoparticles and the 7-MJ nanoparticles an oil-in-water (O/W) emulsion was prepared, whereas for the 7-MJ+RhB nanoparticles a water-in-oil-in-water (W/O/W) emulsion was prepared. For the oil phase of the blank nanoparticles, 50 mg of PLGA 50:50 was dissolved in 5 mL of acetone. For the oil phase of the nanoparticles containing 7-MJ (7-MJ nanoparticles and 7-MJ+RhB nanoparticles), 5 mg of 7-MJ and 50 mg of PLGA 50:50 was dissolved in 5 mL of acetone. For the internal aqueous phase of the 7-MJ+RhB nanoparticles, 1 mL of 2 % RhB was prepared, then added dropwise to its oil phase. The external aqueous continuous phase of each formulation constituted 1% polyvinyl alcohol (PVA). The addition of the above-mentioned oil phase was done in a dropwise manner to the ex-ternal aqueous phase for each formulation. While stirring, one drop of 2 % Tween 80 was added to each formulation as an emulsifier to break the surface tension between the oil and water phase. For each of the nanoformulations the heterogeneous solution was stirred on a magnetic stirring plate for approximately 10 min. The solution was blended by means of a Silverson L4R high speed homogenizer (Silverson Machines Limited, Buckinghamshire, United Kingdom) at 10 000 rpm for 10 min on ice. The emulsion was left to stir for 24 h on a magnetic stirring plate. The resultant emulsion was centrifuged for 30 min at 10 000 rpm to form a nanoparticle pellet. The supernatant was discarded, and the pellet was resuspended in 5 mL of 1 % Trehalose. The re-suspended pellet was rapidly frozen with liquid nitrogen and freeze-dried over a period of 2 days to obtain freeze-dried PLGA nanoformulations (blank nanoparticles, 7-MJ nanoparticles and 7-MJ + RhB nanoparticles).”
To the following “Three PLGA nanoformulations were prepared; blank nanoparticles, 7-MJ nanoparticles, and 7-MJ + Rodamine B (RhB) nanoparticles. The PLGA nanoparticles were prepared through a nanoprecipitation technique. To prepare the blank nanoparticles and 7-MJ nanoparticles, an oil-in-water (O/W) emulsion was used. For the 7-MJ + RhB nanoparticles, a water-in-oil-in-water (W/O/W) emulsion was prepared. For the blank nanoparticles, the oil phase consisted of 50 mg of PLGA (50:50 → LA:GA) dissolved in 5 mL of acetone. For the nanoparticles containing 7-MJ (7-MJ and 7-MJ + RhB), the oil phase included 5 mg of 7-MJ and 50 mg of PLGA (50:50 → LA:GA) dissolved in 5 mL of acetone. For the internal aqueous phase of the 7-MJ + RhB nanoparticles, 1 mL of 2 % RhB was prepared. The RhB solution was added dropwise to the oil phase. The external aqueous continuous phase of each formulation constituted 1% polyvinyl alcohol (PVA). The addition of the above-mentioned oil phase was done in a dropwise manner to the external aqueous phase for each formulation. After the addition, one drop of 2 % Tween 80 was added to each formulation, while stirring on a magnetic stirring plate. The drop of Tween 80 functioned as an emulsifier with the purpose of breaking the surface tension between the oil and water phase. For each of the nanoformulations the heterogeneous solution was stirred for approximately 10 min. The solution was then blended by means of a Silverson L4R high speed homogenizer (Silverson Machines Limited, Buckinghamshire, United Kingdom) at 10 000 rpm for 10 min on ice. Which after the emulsion was left to stir for 24 h. The resultant emulsion was centrifuged for 30 min at 10 000 rpm to form a nanoparticle pellet. The supernatant was discarded, and the pellet was resuspended in 5 mL of 1 % Trehalose. The resuspended pellet was rapidly frozen with liquid nitrogen and freeze-dried over a period of 2 days to obtain freeze-dried PLGA nanoformulations (blank nanoparticles, 7-MJ nanoparticles and 7-MJ + RhB nanoparticles).”
We hope that the revision increased the readability and understanding of preparation of the nanoparticles.
Regarding your suggestion to include an illustration on the synthesis of the nanoparticles. Thank you for your suggestion to include a sketch outlining the procedures for nanoparticle preparation. Diagrams outlining the synthesis of the nanoparticles have been included in the Supplementary Materials section of the manuscript for better clarity and visualization of the procedure. We also included the following to the end of the synthesis section at Line 127-128 “Illustrative diagrams of the synthesis of the PLGA nanoparticles are provided in Figure S1-S3 (Supplementary Materials)”.
We appreciate your valuable input, which has helped enhance the presentation of the PLGA synthesis methodology.
Comment 6: Line 156: Please rename this section to a method for drug analysis. Please make it directly below Chemicals, reagents, and pure compound.
Response 6: Thank you for your valuable suggestion. In response, we have renamed the section of “Ultraviolet-visual spectroscopy” to "Method for drug analysis".
Regarding your suggestion to move the section "Method for drug analysis" directly below the "Chemicals, reagents, and pure compound" section. While we appreciate your input, we believe that placing the drug analysis method before the nanoparticle synthesis and characterization section may not be the most logical structure for the manuscript. The drug analysis method is closely tied to the nanoparticle characterization process, as it analysed the amount of drug incorporated into the nanoparticle. To ensure a clear and logical flow, it is more appropriate for the drug analysis method to be positioned after the description of the nanoparticle synthesis and within the characterization. This allows the reader to first understand how the nanoparticles are synthesized before moving on to the specific methods for characterization of the nanoparticles.
We hope this explanation clarifies the reasoning behind the current structure. We are happy to further discuss any modifications if needed.
Comment 7: Line 159: drug content is usually measured at a fixed wavelength. How drug content was measured between 190 – 800 nm?
Response 7: Thank you for your insightful comment. You are correct that drug content is typically measured at a fixed wavelength. In our study, we performed a wavelength scan between 190 and 800 nm for 7-MJ and RhB at a concentration range of 100 to 1000 μg/mL. This was done too identify the optimal wavelength for measuring the drug content, ensuring that we were detecting the specific absorbance peak of 7-MJ and RhB. The maximum absorbance of 7-MJ and RhB was found at 430 nm and 554 nm, respectively, as mentioned in the results at Line 342-343. We used this wavelength for quantifying the drug content in subsequent analyses.
The following was added at Line 160-161 to help clarify why a broad scan was performed “The broad wavelength scan was performed to identify the optimal wavelength for measuring the drug content of 7-MJ and RhB, respectively.”
We hope this clarifies the methodology used for measuring drug content. Please let us know if any further clarification would be necessary.
Comment 8: Line 162: Please change number for this section to be
2.2.4. Drug loading content (DLC) and Encapsulation efficiency (EE) of the PLGA nanoparticles
Response 8: Thank you for your suggestion. The section “2.2.4.1. Drug Loading Content (DLC) and Encapsulation Efficiency (EE) of the PLGA Nanoparticles” is numbered as 2.2.4.1 because it is a sub-section under “2.2.4. Method for drug analysis”. The section of 2.2.4 addresses the method of scanning the compounds at various concentrations as well as the nanoparticles to obtain the amount of drug in the nanoparticle. The section of 2.2.4.1 specifically covers the calculations of the DLC and EE of the nanoparticles.
We hope this explanation clarifies the structure, and we are happy to consider any further suggestions you may have.
Comment 9: Line 313 – 336: Please try to avoid repetition. Most of the information in the first paragraph is similar to the second paragraph and table. Just give a brief paragraph about the table.
Response 9: Thank you for your helpful comment. We apologize for the repetition in the manuscript. In response, we have revised the section to avoid redundancy. We have condensed the first paragraph, eliminated unnecessary overlap and completely removed the second paragraph.
We removed the wording in bold from the 1st paragraph: “For the determination of the hydrodynamic diameter, polydispersity index (PdI) and zeta potential of the formulated nanoparticles, the dynamic light scattering technique was utilized, with results depicted in Table 1. The formulated nanoparticles depicted a mean hydrodynamic diameter between 250 and 670 nm, PdI between 0.09 and 0.60 and a zeta potential between -14 and -20 mV. Hydrodynamic diameter is an indication of the size of the nanoparticles and PdI is a measurement of the homogeneity of the nanoparticles. Zeta potential is the potential difference between the dispersion medium and the stationary layer of fluid attached to the dispersed particles also known as the electrokinetic potential of the nanoparticles in colloidal dispersions. Therefore, zeta potential indicates the surface charge and potential stability of the nanoparticles in suspension. The mean hydrodynamic diameter of RhB was shown to be, 1763.67 ± 53.15 nm (PdI=0.66 ± 0.12).”
The 2nd paragraph was removed, “The 7-MJ nanoparticles were a bit larger in size with a mean hydrodynamic diameter of 288.27 ± 4.76 nm compared to the blank nanoparticles with a mean hydrodynamic diameter of 257.10 ± 2.97 nm. The 7MJ + RhB nanoparticles had a mean hydrodynamic diameter of 667.30 ± 15.70 nm, The blank nanoparticles and the 7-MJ nanoparticles exhibited PdIs of 0.10 ± 0.009 and 0.09 ± 0.03, respectively, and the 7MJ + RhB nanoparticles exhibited a PdI value of 0.59 ± 0.03. The blank nanoparticles and the 7-MJ nanoparticles both exhibited a zeta potential around -14 mV. Whereas the 7MJ + RhB nanoparticles, exhibited a zeta potential around -20 mV.”.
We believe this revision improves the clarity and flow of the manuscript, and we appreciate your guidance in enhancing its structure.
Comment 10 and 11: Line 361 and 381: If applicable, please make FTIR and XRD scans for pure 7-MJ and RhB.
Response 10 and 11: Thank you for your constructive comment and for suggesting the inclusion of FTIR spectra and XRD curves for pure 7-MJ and RhB. We agree that such analyses would provide valuable insights into the interactions between the polymer and drug.
Therefore, we included the additional FTIR spectra of PLGA, pure 7-MJ and RhB as Figure 1 (B). We also included the additional XRD curves of PLGA, pure 7-MJ and RhB as Figure 2 (B).
We appreciate your valuable input, which has helped improve the clarity of our results.
Comment 12: Line 450: The discussion part is too long and contains some simple principles. Please try to make it simple and attractive to readers.
Response 12: Thank you for your constructive feedback. We understand your concern regarding the length and complexity of the discussion section. In response, we have revised the discussion to make it more concise and focused, eliminating some of the simpler principles and streamlining the content to highlight the most relevant findings and their implications.
We have also worked to improve the readability and flow, aiming to make the section more engaging and accessible to readers. We appreciate your valuable input in helping to enhance the clarity and impact of this section.
Comments on the Quality of English Language:
Some sentences were hard to understand.
Also, use sentences that are shorter.
Response: Thank you for your valuable feedback. We apologize for sentences that were unclear or difficult to understand. In response, we have revised the manuscript to simplify the language and ensure clarity. Additionally, we have shortened several sentences, especially within the discussion section, to improve readability and flow. We hope these revisions make the manuscript more accessible and easier to follow. Please let us know whether further refinement will be required.
Thank you once again for your contribution to this work and for considering our manuscript for publication.

Reviewer 2 Report
Comments and Suggestions for Authors
The manuscript entitled “Antitubercular activity of 7-methyljuglone-loaded poly- (lactide co-glycolide) nanoparticles” developed PLGA nanoparticles to load the natural products for the treatment of tubercular. The study is interesting and provides a nanoplatform to lower the cytotoxicity potential of the natural products. Revisions are necessary prior to the acceptance of the article. Some comments:
1. For the evaluation of drug loading capacity and entrapment efficiency, at least three independent experiments are necessary.
2. It’s better to provide the FTIR spectra and XRD curves of the physical mixture of PLGA and 7-MJ.
3. Some reported on nanomedicines, and drug delivery, such as eBioMedicine 2024, 107, 105301. https://doi.org/10.1016/j.ebiom.2024.105301, may be added to the revised manuscript.
4. For the cytotoxicity study, only the IC50 values are exhibited in Table 4, and it’s better to provide the relative curves.
5. What is the specific mechanism of the toxic side effects of 7-MJ when used in the treatment of tuberculosis? Please describe it briefly.
6. Could 7-MJ be released form the nanoparticles in a sustained and controlled behavior?

Author Response
We would like to sincerely thank you for taking the time to review our manuscript and for providing such thoughtful and constructive feedback. Your detailed comments and suggestions have been invaluable in improving the quality of our work. We greatly appreciate your efforts in considering our study for publication and for contributing to its refinement.
Please see below our response to your comments:
Comment 1: For the evaluation of drug loading capacity and entrapment efficiency, at least three independent experiments are necessary.
Response 1: Thank you for your valuable comment regarding the independent experiments required for evaluating drug loading capacity (DLC) and entrapment efficiency (EE). We agree that adding additional replicates will help to strengthen the statistical robustness of the results.
We have revised the manuscript, accordingly, conducting three independent replicates. We indicated the replicates by taking the average of the triplicates adding the standard deviation thereof.
Comment 2: It’s better to provide the FTIR spectra and XRD curves of the physical mixture of PLGA and 7-MJ.
Response 2: Thank you for your constructive comment and for suggesting the inclusion of FTIR spectra and XRD curves for the physical mixture of PLGA and 7-MJ. We agree that such analyses would provide valuable insights into the interactions between the polymer and drug.
Therefore, we included the additional FTIR spectra of PLGA, pure 7-MJ and RhB as Figure 1 (B). We also included the additional XRD curves of PLGA, pure 7-MJ and RhB as Figure 2 (B).
We sincerely appreciate your suggestion and hope that it strengthens the results of the manuscript.
Comment 3: Some reported on nanomedicines, and drug delivery, such as eBioMedicine 2024, 107, 105301. https://doi.org/10.1016/j.ebiom.2024.105301, may be added to the revised manuscript.
Response 3: Thank you for your thoughtful suggestion to include the article from eBioMedicine 2024, 107, 105301 in our revised manuscript. We appreciate the reference, and we have reviewed the paper as recommended.
We believe that this study is highly relevant to our work and provides valuable insights into nanomedicine and drug delivery. As such, we have incorporated the suggested reference into the manuscript in the Discussion section.
Comment 4: For the cytotoxicity study, only the IC50 values are exhibited in Table 4, and it’s better to provide the relative curves.
Response 4: Thank you for your constructive comment regarding the presentation of cytotoxicity data. We agree that including the dose-response curves alongside the IC50 values would provide a clearer visualization of the dose-response relationship and enhance the interpretation of the cytotoxicity results.
In response to your suggestion, we have now included the dose-response curves in Figure 3, as requested. We believe this addition strengthens the presentation of the data and provides a more comprehensive understanding of the cytotoxic profile of the samples. We appreciate your valuable input, which has helped improve the clarity of our results.
Comment 5: What is the specific mechanism of the toxic side effects of 7-MJ when used in the treatment of tuberculosis? Please describe it briefly.
Response 5: Thank you for your valuable comment and for raising this important point regarding the mechanism of toxic side effects of 7-MJ in the treatment of tuberculosis.
At present, the specific mechanisms underlying the toxic side effects of 7-MJ in TB treatment are not fully elucidated, as this compound is still under investigation. However, based on existing literature and preliminary findings, we hypothesize that the potential toxicity of 7-MJ could involve several mechanisms such as cell apoptosis, oxidative stress, and mitochondrial dysfunction.
For cell apoptosis: 7-MJ has been shown to induce apoptosis in various cell types. In the context of TB, excessive cell death, particularly of immune cells like macrophages, could impair the body’s ability to effectively combat Mycobacterium tuberculosis and result in tissue damage.
For oxidative Stress: Like many bioactive compounds, 7-MJ may promote the generation of reactive oxygen species (ROS), leading to oxidative stress. This could damage cellular components, including lipids, proteins, and DNA, potentially contributing to tissue damage.
For mitochondrial dysfunction: 7-MJ has also been reported to interfere with mitochondrial function, which can impair energy production in cells and lead to apoptosis and systemic toxicity.
We hope this provides some insight into the potential mechanisms. These mechanisms are speculative and based on studies in other contexts, therefore we did not add these to the manuscript. We will however consider adding these mechanisms into the discussion if you think it will add valuable insights to the reader.
Comment 6: Could 7-MJ be released form the nanoparticles in a sustained and controlled behavior?
Response 6: Thank you for your insightful comment regarding the release behavior of 7-MJ from the nanoparticles. The primary objective of our study was to evaluate the uptake of the nanoparticles by macrophage cells, rather than to investigate the release kinetics of 7-MJ.
Therefore, sustained and controlled release studies were not within the scope of this research. However, we acknowledge that investigating the sustained and controlled release would provide valuable information for understanding the therapeutic potential of the formulation. Therefore, this could certainly be a focus of future studies to further evaluate the long-term therapeutic potential of the nanoparticles.
Thank you again for your time and dedication to the review process.

Round 2
Reviewer 1 Report
Comments and Suggestions for Authors
Authors address comments.